# Enhancing Patient Understanding of Perianal Fistula MRI Findings Using ChatGPT: A Randomized, Single Centre Study

**DOI:** 10.3390/diagnostics16010072

**Published:** 2025-12-25

**Authors:** Easan Anand, Itai Ghersin, Gita Lingam, Katie Devlin, Theo Pelly, Daniel Singer, Chris Tomlinson, Robin E. J. Munro, Rachel Capstick, Anna Antoniou, Ailsa L. Hart, Phil Tozer, Kapil Sahnan, Phillip Lung

**Affiliations:** 1Robin Phillips’ Fistula Research Unit, St Mark’s The National Bowel Hospital, London NW10 7NS, UK; itai.ghersin@nhs.net (I.G.); gita.lingam@nhs.net (G.L.); katie.devlin3@nhs.net (K.D.); h.pelly@nhs.net (T.P.); ailsa.hart@nhs.net (A.L.H.); philtozer@nhs.net (P.T.); kapil.sahnan@nhs.net (K.S.); philliplung@nhs.net (P.L.); 2Department of Surgery & Cancer, Imperial College London, London SW7 2AZ, UK; 3Tenrec Analytics, St Albans AL1 4TJ, UK; daniel.singer@tenrecanalytics.com; 4Institute of Health Informatics, University College London, London WC1E 6BT, UK; christopher.tomlinson@ucl.ac.uk; 5St Mark’s The National Bowel Hospital, London NW10 7NS, UKrachelcapstick@hotmail.com (R.C.); anna.antoniou@gmail.com (A.A.)

**Keywords:** artificial intelligence, Crohn’s disease, cryptoglandular fistula, large language models, magnetic resonance imaging, patient communication, perianal fistula

## Abstract

**Background/Objectives:** Large Language Models (LLMs) may help translate complex Magnetic Resonance Imaging (MRI) fistula reports into accessible, patient-friendly summaries. This study evaluated the clinical utility, safety, and patient acceptability of Generative Pre-trained Transformer (GPT-4o) in generating such reports. **Methods:** A three-phase study was conducted at a single centre. Phase I involved prompt engineering and pilot testing of GPT-4o outputs for feasibility. Phase II assessed 250 consecutive MRI fistula reports from September 2024 to November 2024, each reviewed by a multi-disciplinary panel to determine hallucinations and thematic content. Phase III randomised patients to review either a simple or complex fistula case, each containing an original report and an Artificial Intelligence (AI)-generated summary (order randomised, origin blinded), and rate readability, trustworthiness, usefulness and comprehension. **Results:** Sixteen patients participated in Phase I pilot testing. In Phase II, hallucinations occurred in 11% of outputs, with unverified recommendations also identified. In Phase III, 61 patients (mean age 48, 41% female) evaluated paired original and AI-generated summaries. AI summaries scored significantly higher for readability, comprehension, and usefulness than original reports (all *p* < 0.001), with equivalent trust ratings. Mean Flesch-Kincaid scores were markedly higher for AI-generated summaries (66 vs. 26; *p* < 0.001). Clinicians highlighted improved anatomical structuring and accessible language, but emphasised risks of inaccuracies. A revised template incorporating Multi-Disciplinary Team (MDT)-focused action points and a lay summary section was co-developed. **Conclusions:** LLMs can enhance the readability and patient understanding of complex MRI reports but remain limited by hallucinations and inconsistent terminology. Safe implementation requires structured oversight, domain-specific refinement, and clinician validation. Future development should prioritise standardised reporting templates incorporating clinician-approved lay summaries.

## 1. Introduction

Artificial intelligence (AI) is increasingly recognised as a transformative force in healthcare, with large language models (LLMs) like ChatGPT showing particular promise in summarising clinical documents, simplifying complex information, including diagnostic reports, and supporting decision-making [1,2,3,4,5,6,7]. A clear understanding of their disease and its management is crucial for patients, as effective shared decision-making and greater engagement in care have been associated with improved outcomes in chronic conditions such as Inflammatory Bowel Disease (IBD). In perianal Crohn’s disease (pCD), a severe phenotype of Crohn’s disease (CD), complex fistulae can cause substantial social and occupational burden [8]. Management often requires lifelong repeated imaging, with MRI recommended by both European Crohn’s and Colitis Organisation (ECCO) and the European Society for Gastrointestinal and Abdominal Radiology (ESGAR) as the gold standard modality [9,10]. An example MRI of a complex perianal fistula is shown in Figure 1.

However, patient perspectives in radiology remain underexplored despite evidence showing that engaging patients in IBD care improves quality of life and likelihood of remission [11,12]. Our recent global survey of patients’ attitudes to imaging in pCD [13] found that whilst MRI is valued for both accuracy and insight, standard MRI fistula reporting is often complex and difficult to understand. Patients expressed a strong interest in AI-generated patient-friendly summaries with actionable recommendations, provided these are clearly explained and validated by professionals [13].

MRI reports are traditionally written for clinicians using technical terminology and assume a medical background. With imaging now routinely digitised and reports increasingly available to patients without a detailed lay person explanation of its contents, LLMs offer a compelling opportunity to automate care pathways and generate personalised, patient-friendly summaries. Patients and families consistently express a strong desire for AI in healthcare, particularly when it enhances the readability, accessibility, and personal relevance of medical information [13,14,15,16,17,18,19]. Studies show high acceptability and even preference for AI-generated content, provided it is trustworthy and context-specific [17,18]. In parallel, medical institutions and specialist societies are increasingly exploring responsible integration of AI, calling for structured data integration, governance frameworks, and alignment with clinical workflows [20,21,22,23]. Specialist taskforces including the ACPGBI AI & Data Management Taskforce and ECCO’s 9th Scientific workshop in AI have been established to support ethical implementation and build clinician and public trust [24,25,26].

Radiology research into the application of LLMs has shown promise in supporting clinical decision making by aligning with imaging appropriateness criteria [27], enhancing accessibility through generating patient-friendly summaries [2,28] and simplified terminology [1,3,29], structuring unformatted reports [30] and supporting follow-up tracking [31]. Yet despite this momentum, real-world evidence on the feasibility, benefits, and risks of AI, particularly from the patient and frontline clinician perspective, remains limited [32,33]. Furthermore, concerns persist around factual inaccuracies, oversimplification, and hallucinations, highlighting the need for expert oversight and further validation before routine clinical use [34,35,36].

This randomised feasibility study aims to evaluate the use of a general-purpose LLM (ChatGPT-4o) to generate lay summaries of MRI reports for patients with complex perianal fistulae. The primary objective is to assess patient-perceived comprehensibility, readability, and usefulness. Secondary objectives include a multi-disciplinary clinician evaluation of factual accuracy, completeness, and the presence of hallucinations or misleading content.

## 2. Materials and Methods

### 2.1. Patient and Public Involvement (PPI)

A dedicated PPI workshop and global survey informed our study’s priorities [13]. Patients with complex perianal fistulising disease advocated for AI-generated summaries that were clear, comprehensible, and actionable. Three patient advocates joined the study steering group and helped design the evaluation framework. Their input shaped the prompt structure, report content preferences, and interpretation of findings.

### 2.2. Study Design and Overview

This was a mixed-methods study evaluating the feasibility, accuracy, and patient perception of large language model (LLM)-generated summaries of MRI reports for benign complex perianal fistulising disease. The study was conducted in three phases:Prompt engineering and pilot testing (results published in the Journal of Imaging [37])Clinical evaluation of LLM outputsPatient evaluation: Randomised patient study assessing readability, comprehensibility, perceived utility, trustworthiness and follow-up questions.

The study used While not fine-tuned for radiology or perianal disease, it draws on a broad dataset including open access medical literature up to August 2023 and has undergone reinforcement learning for safety and alignment. Outputs were reviewed by clinicians before patient use. As GPT-4o lacks DICOM training and regulatory approval, its use was limited to supporting patient communication within a research context and not clinical decision-making.

### 2.3. Setting and Participants

A total of 250 consecutive, de-identified MRI reports were collected from adult patients (≥18 years) undergoing imaging for benign fistula-in-ano conditions at a UK tertiary centre. Reports were included if they described fistulae related to inflammatory bowel disease, cryptoglandular origin, or ileoanal pouch surgery. Exclusion criteria included patients < 18 years, malignancy or absence of a fistula.

### 2.4. Phase 1: Prompt Engineering & Pilot Phase

A pilot study involving sixteen patients with fistulae was conducted to optimise prompts and study design, and to determine the appropriate sample size. A brief report of the pilot can be found at the Journal of Imaging [37].

Prompt engineering was iteratively refined to guide GPT-4o in generating patient-friendly MRI summaries of anorectal fistula reports. Prompts included relevant, anonymised, clinical history and imaging findings, followed by clear instructions to simplify terminology to a ~12-year-old reading level (Flesch-Kincaid ≥ 60), use clock-face analogies to describe fistula location, and provide actionable, patient-tailored recommendations (based on qualitative feedback from a PPI day). Variations in phrasing were tested using clinician feedback to optimise accuracy and clarity. GPT-4o used structured radiology text as input and was not fine-tuned on clinical data. All outputs were reviewed by clinicians before being shared with patients. Outcomes from the pilot phase were utilised to determine an appropriate sample size for the primary patient evaluation study.

#### Sample Size Calculation

Power calculations for the primary outcomes were performed to estimate the number of participants required to achieve 90% power (α = 0.05) in the main phase of the study. Effect sizes were derived from pilot data: readability (d = 1.08), comprehensibility (d = 1.11), utility (d = 0.92), follow-up intentions (d = 0.82), and trustworthiness (d = 0.61). Based on these estimates, required sample sizes per arm ranged from 9 participants for readability and comprehensibility to 29 participants for trustworthiness. To ensure sufficient power across all outcomes, a conservative sample size of 30 participants per arm was targeted.

### 2.5. Phase 2: Clinical Evaluation of AI Summaries

Each of the 250 AI-generated summaries was independently reviewed by at least two clinicians from a 6-team multidisciplinary panel (2 radiologists, 2 colorectal surgeons, 2 gastroenterologists). Fifty cases underwent assessment by all 6 readers and case-weighted means were calculated. Discrepancies were resolved through consensus discussion.

All assessors received training with structured templates and clear annotation guidelines. The evaluation framework excluded comments on AI competence to ensure unbiased assessment.

Evaluation criteria included:•Fidelity to original report: ○Yes○No, but not clinically significant○No, clinically significant (hallucination)•Quantitative ratings (1–5): ○Overall impression○Strength of recommendations○Confidence in report•Hallucination detection: ○Presence (HS harm scale [38]: none, mild, moderate, severe). Hallucinations were classified according to the WHO/NHS harm-severity framework, using the International Classification for Patient Safety categories (none, mild, moderate, severe, death) to assess potential patient-impact.

### 2.6. Phase 3: Randomised Patient Evaluation

Patients were randomised using a computer-generated sequence to review one original and one AI-generated summary of the same report, matched to simple or complex cases. Both the order of presentation and the report type were blinded.

Patients scored reports on a 5-point Likert scale (1 = strongly disagree, 5 = strongly agree) across the following domains:•Readability•Comprehensibility•Perceived utility•Likelihood of follow-up questions•Trustworthiness

Qualitative feedback was also collected and analysed thematically to explore comprehension, preferences, and suggestions for improvement.

### 2.7. Data Analysis

#### 2.7.1. Quantitative Analysis

Ratings were analysed using either means ± standard deviation (SD) or medians with interquartile range (IQR), depending on data distribution. Paired *t*-tests were used for normally distributed data; non-parametric tests (e.g., Wilcoxon signed-rank) were applied for skewed data. Analyses were conducted in IBM SPSS Statistics v29.

#### 2.7.2. Qualitative Analysis

Open-text feedback from patients and clinicians was thematically coded and analysed using Braun & Clark methodology [39]. This informed prompt refinement and highlighted specific strengths, areas for improvement, and potential for harm in AI outputs.

### 2.8. Ethical Approval and Data Availability

The study received ethical approval from the Research Ethics Committee (REC Reference: 24/WA/0374), with institutional approval from London North West University Healthcare NHS Trust.

GPT-4o model architecture and weights remain proprietary to OpenAI.

## 3. Results

### 3.1. Phase 1: Prompt Engineering & Pilot Phase

A pilot study involving 16 patients recruited from a tertiary outpatient setting in a single centre evaluated the feasibility of AI-generated MRI fistula summaries, which were designed for a reading age of 12 [37]. In a blinded, randomised comparison, AI-generated summaries were expectedly rated significantly higher than original reports (written for clinicians) for readability (median 5 vs. 2, *p* = 0.011), comprehensibility (median 5 vs. 2, *p* = 0.007), and overall quality (median 4.5 vs. 4, *p* = 0.013). Patients were also less likely to have follow-up questions after reading AI summaries (median 3 vs. 4, *p* = 0.018), though both report types were rated similarly for trustworthiness. Clinician assessments confirmed these findings, with AI summaries achieving higher Flesch-Kincaid scores (mean 54.6 vs. 32.2, *p* = 0.005), full marks for quality, and no hallucinations or critical errors. The iterative refinement of the prompt has been fully described in our original pilot study, published in the Journal of Imaging [37], which details all earlier prompt versions and the stepwise modifications applied. For the present study, we incorporated both the quantitative results and qualitative patient feedback from that pilot, alongside input from patient representatives within the study group, to develop the final prompt used (Figure 2). Key refinements focused on elements patients identified as most helpful—simple language, clearer anatomical descriptions including clock-face positions, structured actionable recommendations, and improved formatting. The Flesch–Kincaid target of ≥60 was unchanged; the higher mean score observed in the current study (65.83) reflects natural variation rather than a change in the prompt specification. AI-generated summaries were produced rapidly, in an average of 18.2 s per report. An example AI-generated summary is provided in Figure 3.

### 3.2. Phase 2: Clinician Evaluation of 250 Consecutive MRI Fistula Reports

A study flowchart for Phase 2 is provided in Figure 4.

Table 1 shows the aetiology and anatomical classification of the 250 consecutive MRI fistula scans analysed during Phase II of the study.

Among 250 consecutive MRI reports, the most common fistula aetiologies were cryptoglandular (63.2%), Crohn’s disease (27.6%), pouch-related (7.6%), and obstetric-injury related rectovaginal fistulae (1.6%). The majority of fistulae were transsphincteric (61.2%), with intersphincteric (25.2%), extrasphincteric (6.8%), subsphincteric (3.6%), and suprasphincteric (3.2%) tracts less frequent. Single fistula tracts were reported in 46.0% of cases, two tracts in 32.0%, and three or more in 18.8%. Horseshoe extensions and abscesses were identified in 15.6% of scans, and 27.6% showed additional fistula extensions beyond the primary tract.

Objective clinician assessment of 250 MRI reports showed significant differences in readability metrics between original and AI-generated summaries (Table 2). AI reports had a much higher Flesch-Kincaid readability score (65.83 vs. 26.25; *p* < 0.001), indicating easier readability, and a lower FK Grade level (7.93 vs. 14.1; *p* < 0.001), suggesting they were accessible to a broader audience. Across all 250 cases, clinicians reported that the AI-generated summaries reflected the original radiology report in 212/250 cases (84.8%), with discrepancies in the remaining 38/250 cases (15.2%). The overall impression of the summaries was high, with a case-weighted mean score of 4.33 on a 5-point scale. Clinicians rated the strength of recommendations at a mean of 4.39, and confidence in the reports at a mean of 4.68. Hallucinations were identified in 29/250 cases (11.6%) and were typically found in aberrant clock-face descriptions. Overall, 2.4% cases were considered as causing mild harm primarily due to minor discrepancies in anatomical or descriptive details. These errors or hallucinations were considered mildly harmful as they could potentially cause patient anxiety or confusion, e.g., if an AI-generated summary incorrectly indicated the side of a fistula and, in a worst-case scenario, could mislead clinicians if they relied solely on the AI-generated summary. No hallucinations were classified as causing moderate or severe harm, as standard clinical practice would require clinicians to verify AI-generated summaries before making any patient care decisions.

### 3.3. Phase 3: Patient Evaluation of AI-Generated Summaries

Figure 5 depicts the recruitment of patients to the patient-evaluation arm of the study (Phase 3).

Table 3 presents demographic data for the 61 patients recruited to Phase 3 of the study. The mean age was 48 years (SD 13.7), and 59% were male. Most participants identified as White (70.5%), followed by Asian (18.0%), Arab (9.8%), and Mixed ethnicity (1.6%). Educational attainment was diverse, with 31.1% holding a university bachelor’s degree and 26.2% having completed graduate or professional qualifications. The majority of participants reported either fluent (68.9%) or advanced (26.2%) proficiency in English reading and writing, while 4.9% reported intermediate proficiency. The most common fistula aetiology was cryptoglandular disease (49.2%), followed by Crohn’s disease (37.7%).

Most participants (85.2%) reported using the NHS app, with 32.8% accessing their records a few times a year and 32.8% a few times a month (Table 4). Despite this, 39.4% found diagnostic reports difficult to understand. The majority found a patient-friendly AI-generated summary (78.7%) and a clear ‘next steps’ or action plan (82.0%) to be extremely or very useful.

### 3.4. Patient Evaluation of Original Reports vs. AI-Generated Patient Summaries

Sixty-one patients were randomised to review either a pair of simple or complex MRI fistula reports, comprising the original radiology report and an AI-generated patient-friendly summary (Table 5). Patients were blinded to the complexity of the case and the origin of the reports (AI or original), and the order of report presentation was randomised. Across both arms, AI-generated summaries were rated significantly higher than original reports or readability, comprehensibility, and perceived utility. This improvement remained statistically significant (*p* < 0.001) even after adjusting for report order, report complexity, and participant characteristics. In the simple fistula group (*n* = 31), AI summaries demonstrated a marked increase in readability (mean difference = 1.19, *p* < 0.001), comprehensibility (1.19, *p* < 0.001), and perceived utility (0.84, *p* < 0.001). Follow-up questions were slightly reduced in the AI group, but this was not statistically significant (*p* = 0.19). Trustworthiness ratings were equivalent between the two formats (*p* = 0.23). In the complex fistula group (*n* = 30), the differences were even more pronounced. AI summaries were significantly more readable (mean difference = 1.80, *p* < 0.001), comprehensible (1.60, *p* < 0.001), and perceived as more useful (1.43, *p* < 0.001) than original reports. There was a non-significant trend toward fewer follow-up questions after reading the AI summaries (*p* = 0.14), trustworthiness again remained equivalent (*p* = 0.36). There was no statistically significant difference in subjective metrics between the AI reports generated in Option A vs. Option B. Patients were able to correctly tell a report was AI-generated 74% of the time when the report was simple, and 62% of the time when it was a complex MRI fistula report.

Patient and clinicians generally viewed the AI-generated MRI summaries as clear, logically structured, and more accessible than standard reports (Table 6). All assessors appreciated formatting features like bullet points and spacing, as well as accurate anatomical localisation using clockface notation when correctly applied. Clinicians highlighted improved readability and satisfactory explanation of complex fistula anatomy, including tract extensions and post-surgical changes. The “action plan” section was valued for its practical relevance and multidisciplinary framing. However, concerns were raised about variable anatomical precision, occasional errors in localisation or omitted findings including vague language (e.g., “travels through the muscles”) and incorrect clockface references. Clinicians flagged hallucinated references to Crohn’s activity (in cases of cryptoglandular fistulas) and missing incidental findings (e.g., hernias and pelvic lumps unrelated to the fistula tract). There was concern regarding overly generic or minimising language around surgical procedures, insufficient detail for complex planning, and the need for clearer, more patient-specific recommendations. In response, a structured reporting template was developed incorporating the most positively received elements, including an optional MDT section and lay AI-generated summary (Table 7).

## 4. Discussion

### 4.1. General Findings

This study was shaped by our Patient and Public Involvement (PPI) day, where patients with pCD emphasised the need for MRI reports written for lay persons, whilst also containing actionable guidance [13]. These priorities echo wider evidence that meaningful engagement improves comprehension and outcomes in chronic disease [11,12]. Building on this, we conducted a three-phase patient- and clinician-led study to assess AI-generated lay summaries of MRI reports in complex perianal fistula disease: a pilot with 16 patients to refine AI prompts; a case series of 250 reports evaluated by clinicians for accuracy, safety, and readability; and a randomised study of 61 patients comparing original versus AI-generated reports using structured Likert scales.

Across all phases, AI-generated summaries, which were specifically designed to be read by a patient with a 12-year-old reading level, were consistently rated higher (*p* < 0.001) for readability, comprehensibility, and overall usefulness, with patients reporting fewer follow-up questions and equal levels of trust compared to original reports, which are written for clinicians. Patients and clinicians attributed the improved readability of AI-generated lay summaries to their structured layout, simplified language (supported by objective improvements in Flesch–Kincaid scores), and clearer anatomical descriptions. Patients and clinicians particularly valued the clarity, structure (including bullet points which are not always available in radiology reporting software), and practical clinical framing of the summaries, although clinicians repeatedly raised concerns about oversimplification and occasional anatomical imprecision. These findings highlight the potential of LLM tools to enhance patient-centred communication in a highly complex disease area, where even the most experienced clinicians can struggle to convey salient findings to patients, but this must be balanced against the inherent nature of existing LLMs to produce hallucinations, defined as factually incorrect content.

Attempts to provide a visual description of the clock face of fistula anatomy were generally well received by patients (although they were unable to verify its accuracy) but some generic recommendations risked being misleading when applied across heterogeneous groups, highlighting some of the pitfalls of a generic transformer-based LLM. For instance, cryptoglandular fistula patients were given an option to follow up with an IBD team, and pouch fistula patients, without context on pouch indication, were presented with generic and IBD-related advice, which could be inaccurate for those with familial adenomatous polyposis. These are not hallucinations per se but represent potentially confusing statements for patients. Conversely, positive examples included translating radiological improvements, such as reduced fistula volume and clearer anatomy, into actionable advice (Table 8).

Hallucinations were identified in up to 11% by clinicians, a quarter of which were judged to be mildly harmful. These findings are relatively high compared with external benchmarks. For example, the Vectara (an AI company) leaderboard [41], which applies a narrow and consistent definition of hallucination across models, reports a typical hallucination rate for GPT-4o of ~1.5%. Although our methodology is not directly comparable, this discrepancy raises important questions. The higher hallucination rate likely reflects the greater difficulty and precision required in medical imaging summarisation, where small anatomical inaccuracies are counted as errors. Hallucination rates in radiology-focused LLM studies are typically higher than in general testing, with one study reporting rates of 6% [42] whilst studies investigating errors in the assessment of medical literature have reported rates as high as 40% [43]. It might indicate that creating technically accurate medical summaries is harder than general-domain tasks because there aren’t enough medical reports in the training data, or alternatively, that clinical evaluators apply a higher standard of scrutiny than benchmark datasets. Hallucinations increased from the pilot study [37] (no hallucinations) to the full study when prompts involved more complex tasks, such as anatomical localisation using clockface notation. These types of errors, though infrequent, raise safety concerns, especially if summaries are delivered without clinical validation. Moreover, the subjective reliability of LLM output is highly sensitive to prompt structure and phrasing, which requires expertise in prompt engineering and introduces further variability. Transformer-based models such as Chat-GPT are inherently stochastic, resulting in non-deterministic (or probabilistic) outputs. Ensuring consistency and reproducibility requires strict constraints and this may be achieved in subsequent iterations with more refined prompts and tailoring of information, including a visual and anatomical description of fistula anatomy and clockface, for example.

Whilst actionable recommendations were consistently highlighted as important by patients, generating accurate, personal statements without access to the full medical record remains a major challenge for generic LLMs. Nonetheless, our findings suggest that concise two- to three-line recommendations at the end of a report could enhance patient engagement, which is itself linked to improved outcomes. Patients, despite being blinded to report type, placed equal trust in AI-generated summaries and radiologist reports, consistent with broader evidence of patient support for AI in healthcare [14,15,16,17,18]. However, this trust reflects confidence in the report as presented, not awareness of potential inaccuracies. If errors were recognised, trust would likely decline, underscoring the need for validation and transparent communication to ensure safety.

A further practical consideration is the potential for LLMs to inadvertently increase workload rather than alleviate it. While designed to streamline communication, generated content may require time-consuming review and approval by clinicians, particularly given the 2 times increase in word count for AI-generated reports, adding administrative burden. This is particularly relevant in radiology, where turnaround times and medico-legal responsibility are already tightly constrained. If poorly implemented, the use of LLMs could compromise documentation quality or reduce the clarity and efficiency of clinical workflows, as noted in other healthcare contexts [33]. Lastly, the automatic insertion of AI-generated lay summaries into patient portals could erode patient-clinician trust if inaccuracies remain uncorrected or cause confusion.

### 4.2. Study Limitations

This study did not address several critical systemic issues that were beyond the scope of this feasibility study, such as comparison with competing LLM models (e.g., DeepSeek, Gemini and CoPilot), cost-effectiveness of AI, infrastructure burden, or environmental impact of deploying LLMs at scale in healthcare settings. These are non-trivial concerns, particularly given the substantial energy requirements and ongoing need for human oversight to validate output. Time constraints meant that certain aspects, such as expanding the breadth of report types and analyses, could not be fully explored and should be addressed in future work. ChatGPT was selected due to its accessibility, widespread use, and robust language generation capabilities at the time of study design. We acknowledge that other LLMs (e.g., Gemini, DeepSeek, Sider, Claude) may produce different outputs, and model performance may vary depending on prompting and domain expertise. Future work could compare multiple models to determine relative reliability, but in clinical contexts, trust must always be mediated by expert review rather than reliance on any single AI model.

Although efforts were made to recruit a broad sample of readers, this could be widened further to capture greater diversity. The patient sample in Phase 3 was predominantly White (70.5%) and highly educated (over 57% with university degrees), which may limit the generalizability of findings to more diverse populations or those with lower health literacy. Future studies should aim to include broader, multi-centre cohorts to ensure equity and inclusivity in AI-assisted patient communication research. Additionally, each patient reviewed a small number of AI-generated reports as participants themselves indicated limited willingness to read multiple reports in one sitting, although this was partially mitigated by recruiting a larger cohort. A more comprehensive assessment of trustworthiness and accuracy could be achieved by providing patients with their own fistula reports and AI-generated summaries and presents an intriguing opportunity for future research. From a regulatory and ethical standpoint, challenges remain around transparency, data governance, and fairness. Many widely used LLMs are opaque in terms of their training data and algorithmic processes, limiting clinician and patient trust [44]. Risks of bias, especially in underserved populations, persist due to non-representative training datasets [32,45]. Furthermore, automation bias means the tendency of clinicians to over-rely on AI outputs even when incorrect has been widely reported and may degrade diagnostic vigilance over time [46]. This is particularly pertinent when one considers that radiologists are already time-limited and unable to produce additional lay summaries, let alone verify the accuracy of AI-generated summaries.

### 4.3. Clinical Implications and Future Directions

This study reinforces that AI-generated summaries offer value, but cannot yet replace radiologist-authored reports, particularly in complex, high-stakes cases. Future development should focus on bespoke, clinically fine-tuned LLMs trained on high-quality, domain-specific datasets and validated against gold-standard clinical benchmarks. Off-the-shelf, general-purpose models lack precise anatomical language or contextualised clinical reasoning. Tailored models, ideally co-developed by interdisciplinary teams of clinicians, data scientists, and patients, could help mitigate hallucinations and improve the relevance and safety of AI-generated content. Progress will also rely on specialised biomedical models, like the open-source BiomedGPT, which are designed to remain stable and avoid losing previously learned knowledge when tackling complex medical tasks [47,48]. BiomedGPT has shown satisfactory performance, with an 8.3% error rate in generating complex radiology reports [47] but further work is clearly required to lower rates of hallucinations and errors to an acceptable level. Specific features desired by patients [13] such as accurate clockface annotations, clear descriptions of disease trajectory, and simplified anatomical diagrams, which are currently beyond the scope of open source models, hold promise if reliably automated. Importantly, AI-generated summaries could help reduce patient anxiety and residual uncertainty by translating complex radiological findings into understandable, actionable language. The rapid evolution of digital health, particularly the Internet of Things (IoT), may enable real-time symptom tracking and delivery of patient-friendly MRI summaries in the future. Emerging 3D-printing techniques can translate complex fistula anatomy into tangible models to enhance patient understanding and support shared decision-making [49,50]. Iterative development of LLMs grounded in robust clinical knowledge and ethical implementation could transform these systems into essential tools that enhance patient engagement and, as Topol [51] emphasised, the ultimate promise of AI lies not in replacing the clinician, but in “deepening the human connection in medicine” by offloading routine documentation and facilitating more meaningful interactions.

## 5. Conclusions

In conclusion, AI-generated MRI summaries can improve patient communication by enhancing readability, structure, and accessibility, particularly in complex conditions requiring repeated imaging. While not yet suitable for standalone use due to inaccuracies and limited contextual nuance, they have potential as valuable adjuncts within radiology workflows, supporting standardised reporting, streamlining documentation, and providing clinician-validated lay summaries. Generative AI models such as ChatGPT can produce coherent and patient-friendly summaries; however, as our study demonstrates, they are prone to inaccuracies (hallucinations) and omissions. Therefore, outputs must be interpreted cautiously and always verified by clinicians before informing patient care. Our study focused on evaluating feasibility, readability, and potential patient comprehension rather than clinical decision-making. Patients value simplified language and actionable guidance in their reports, which we have incorporated. Safe integration requires rigorous clinical oversight, domain-specific model refinement, and ethical safeguards that prioritise patient safety, equity, and trust. Future work should focus on standardised structured reporting templates, clinician-validated AI summaries, expansion into other radiology subspecialties and medical disciplines, all with continued patient input and approval.

## Figures and Tables

**Figure 1 diagnostics-16-00072-f001:**
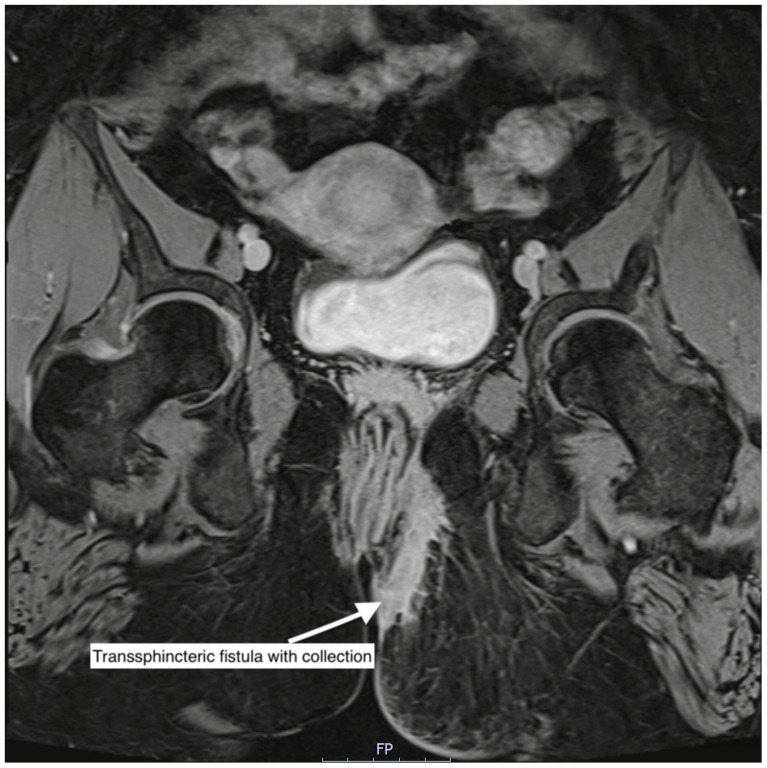
Coronal view of a complex transsphincteric fistula with collection in perianal fistulising Crohn’s disease.

**Figure 2 diagnostics-16-00072-f002:**
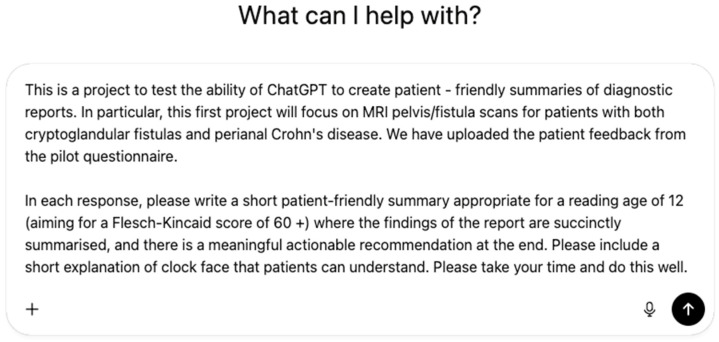
Final Prompt based on iterative feedback.

**Figure 3 diagnostics-16-00072-f003:**
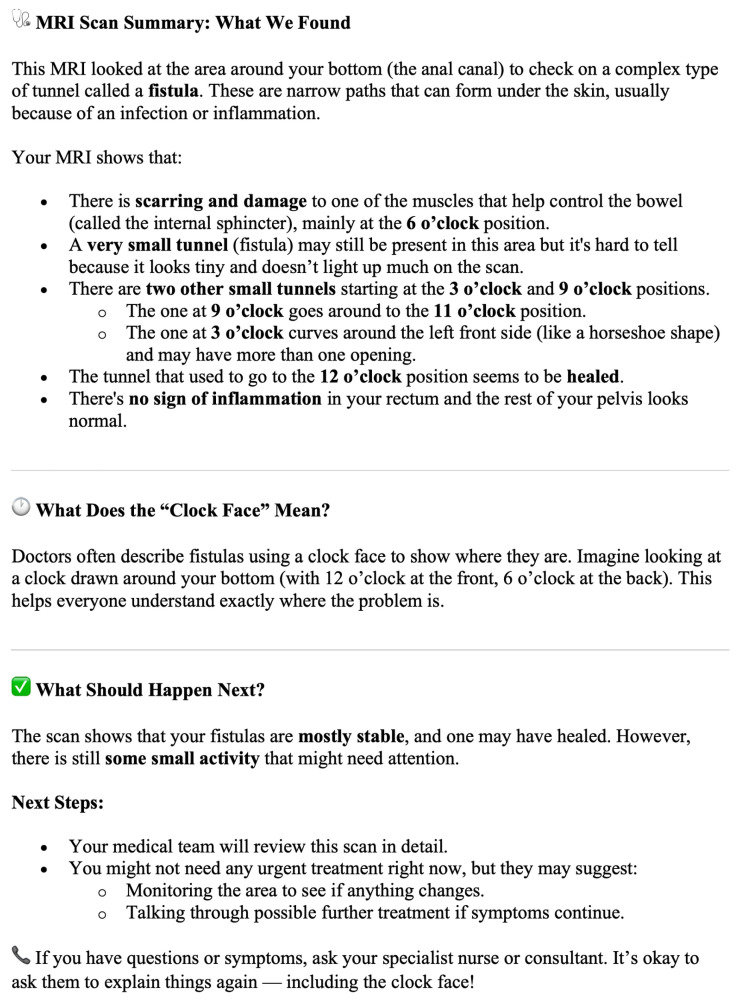
Example AI-generated Summary.

**Figure 4 diagnostics-16-00072-f004:**
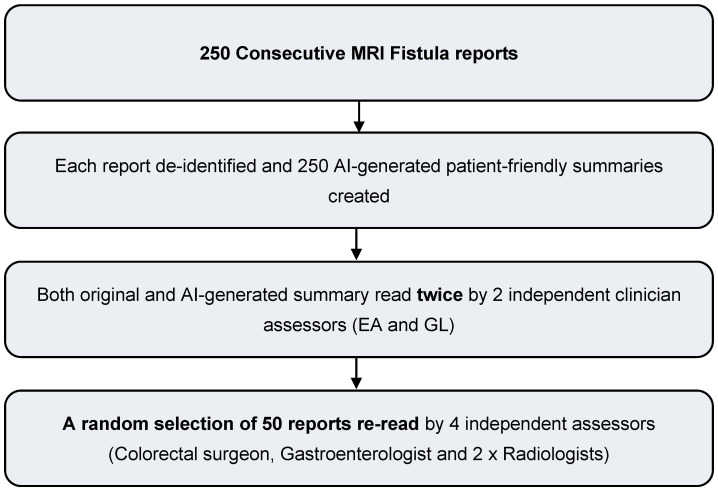
Clinician evaluation workflow.

**Figure 5 diagnostics-16-00072-f005:**
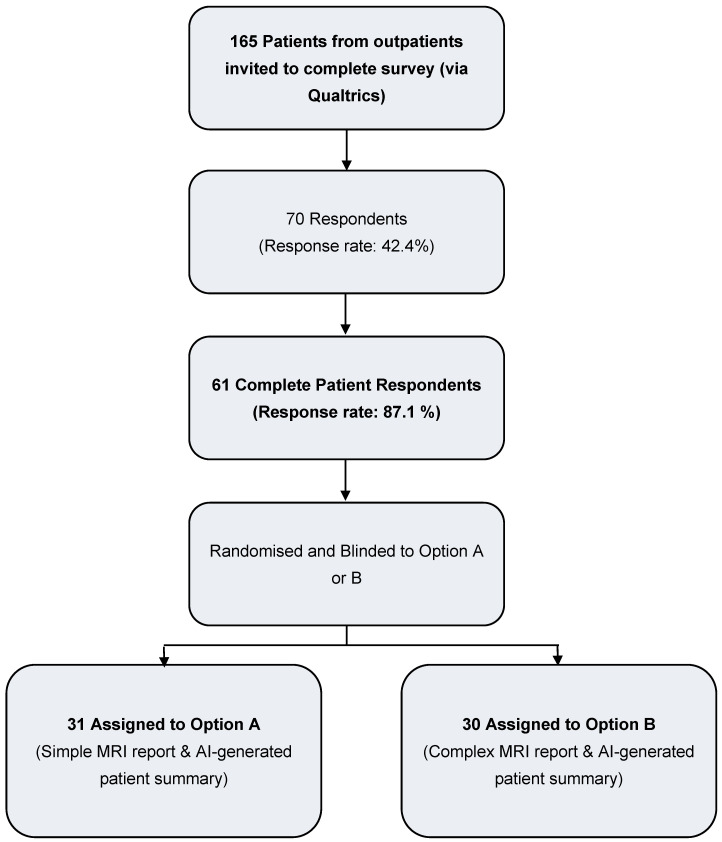
Patient Study Flowchart.

**Table 1 diagnostics-16-00072-t001:** Fistula characteristics on MRI reports (*n* = 250).

Characteristic	Category	Frequency	Percent (%)
Aetiology of fistulae	Cryptoglandular	158	63.2
	Crohn’s	69	27.6
	Pouch-related fistula	19	7.6
	Rectovaginal fistula	4	1.6
Parks’ classification	Subsphincteric	9	3.6
	Intersphincteric	63	25.2
	Transsphincteric	153	61.2
	Suprasphincteric	8	3.2
	Extrasphincteric	17	6.8
Horseshoe	No horseshoe	211	84.4
	Horseshoe collection	39	15.6
Extension	No extension	181	72.4
	Extension	69	27.6
Abscess	No abscess	211	84.4
	Abscess	39	15.6

**Table 2 diagnostics-16-00072-t002:** Objective Metrics & Clinician Evaluation.

Objective Metric	Original (Mean (SD))	AI-Generated Summary (Mean (SD))	t	*p*
Flesch-Kincaid Score	26.25 (9.1)	65.83 (5.0)	58.243	<0.001
FK Grade	14.1 (2.7)	7.93 (1.03)	−31.92	<0.001
Word Count	172 (57.35)	345 (55.03)	48.52	<0.001
**Subjective Metric**	**Combined Clinician assessment**			
AI reflects original report	Yes: 212/250 (84.8%) No, but not clinically significant ^+^: 38/250 (15.2%)			
Overall impression (1–5)	4.33 *			
Strength of recommendations (1–5) *	4.39 *			
Confidence in report (1–5) *	4.68 *			
Hallucinations present	No: 221/250 (88.4%) Yes: 29/250 (11.6%)			
Severity of hallucinations	2.4% Cases considered as ‘mild harm’			

Table 2 highlights objective and subjective metrics analysed during Phase II of the study. Mean (SD) scores are provided for normally distributed objective metrics. Subjective metrics were assessed by at least 2 readers for all 250 scans. 50 scans were assessed by 6 readers. * Case-weighted means for subjective metrics are calculated based on per-case average of 6 readers (2 Surgeons, 2 Gastroenterologists, 2 Radiologists) for the first 50 scans, and 2 readers (2 Surgeons) for the remaining 200 scans. ^+^ Omissions were defined as ‘not clinically significant’ if they were incidental findings unrelated to fistula tract, anatomy or disease e.g., incidental gallstone finding on combined MR enterography.

**Table 3 diagnostics-16-00072-t003:** Patient Reader Demographics & Disease Characteristics (*n* = 61).

Patient Demographics	Mean (SD)/*n* (%)
Age	
Mean (SD)	48 (13.7)
**Sex**	
Male	36 (59%)
Female	25 (41%)
**Ethnicity**	
Asian	11 (18.0%)
Mixed	1 (1.6%)
Arab	6 (9.8%)
White	43 (70.5%)
**Level of Education**	
Primary School	1 (1.7%)
Secondary School	12 (19.7%)
Vocational or similar	13 (21.3%)
University bachelor’s degree	19 (31.1%)
Graduate or professional	16 (26.2%)
**English—Reading and Writing**	
Fluent (near-native proficiency)	42 (68.9%)
Advanced (comfortable with complex reading and writing)	16 (26.2%)
Intermediate (can read and write but with some difficulty)	3 (4.9%)
**Fistula Aetiology**	
Crohn’s Disease	23 (37.7%)
Cryptoglandular Disease	30 (49.2%)
Ulcerative Colitis	1 (1.6%)
Pouch-related Fistula	3 (4.9%)
Rectovaginal Fistula	4 (6.6%)
Number of investigations since fistula diagnosis	Mean (S.D.)
MRI Fistula	6.3 (6.1) Add Range

Table 3 highlights the demographic data for the 61 patients who participated in Phase 3 of the Study. Specific and relevant characteristics such as ethnicity, education level and reading proficiency are presented in this table, as well as individual patient fistula aetiology to highlight the broad patient-base.

**Table 4 diagnostics-16-00072-t004:** Use of NHS Digital Services (*n* = 61).

Question	Response Category	*n* (%)
Do you currently use the NHS app?	Yes	52 (85.2%)
	No	9 (14.8%)
How often do you access your medical records on this app?	Never	3 (4.9%)
	Once a month	7 (11.5%)
	A few times a year	20 (32.8%)
	Prior to clinical appointments	11 (18%)
	A few times a month	20 (32.8%)
How easy do you find it to read your diagnostic reports?	Extremely difficult	7 (11.5%)
	Slightly difficult	17 (27.9%)
	Neither easy nor difficult	17 (27.9%)
	Slightly easy	11 (18.0%)
	Extremely easy	9 (14.8%)
How useful would you find a patient-friendly AI-generated summary of your diagnostic report?	Extremely useful	28 (45.9%)
	Very useful	20 (32.8%)
	Moderately useful	9 (14.8%)
	Slightly useful	3 (4.9%)
	Not at all useful	1 (1.6%)
How useful would you find a ‘next steps’ or ‘action plan’ statement?	Extremely useful	32 (52.5%)
	Very useful	18 (29.5%)
	Moderately useful	8 (13.1%)
	Slightly useful	2 (3.3%)
	Not at all useful	1 (1.6%)

Table 4 summarises patients’ responses regarding digital health literacy, use of electronic health records and interfaces, and their preferences for patient-friendly summaries.

**Table 5 diagnostics-16-00072-t005:** Patient Evaluation of Original vs. AI Report summaries.

Report A—Simple Fistula (*n* = 31).					
Variable	Mean Original (SD)	Mean AI (SD)	Mean Difference	t	*p*-Value
Readability	3.26 (1.21)	4.45 (1.03)	1.19	4.27	**<0.001**
Comprehensibility	3.29 (1.10)	4.48 (0.93)	1.19	5.21	**<0.001**
Perceived Utility	3.45 (1.15)	4.29 (0.97)	0.84	3.47	**<0.001**
Follow Up	4.19 (0.95)	4.00 (1.18)	−0.19	−0.90	0.19
Trustworthiness	4.06 (1.00)	4.23 (1.12)	0.16	0.740	0.23
**Report B—Complex Fistula (*n* = 30)**					
Variable	Mean Original (SD)	Mean AI (SD)	Mean Difference	t	*p*-Value
Readability	2.70 (1.26)	4.50 (0.94)	1.80	5.5	<0.001
Comprehensibility	2.73 (1.29)	4.33 (1.16)	1.60	4.7	<0.001
Perceived Utility	2.93 (1.14)	4.37 (1.16)	1.43	5.0	<0.001
Follow Up	4.23 (0.97)	3.73(1.20)	−0.50	−1.53	0.14
Trustworthiness	3.97 (0.93)	4.13 (1.11)	0.17	0.93	0.36
**Comparison of AI Across Differing Complexities (Option A—AI vs. Option B—AI)**					
Variable	A: Mean AI (SD)	B: Mean AI (SD)	Mean Difference	t	*p*-Value
Readability	4.45 (1.03)	4.50 (0.94)	−0.05	−0.19	0.85
Comprehensibility	4.48 (0.93)	4.33 (1.16)	0.15	0.56	0.58
Perceived Utility	4.29 (0.97)	4.37 (1.16)	−0.08	−0.28	0.78
Follow Up	4.00 (1.18)	3.73 (1.20)	0.27	0.87	0.39
Trustworthiness	4.23 (1.12)	4.13 (1.11)	0.09	0.33	0.75

Table 5 presents Phase 3 results of subjective assessments of readability, comprehensibility, perceived utility, follow-up questions, and trustworthiness across three subgroups. Subgroup 1: mean (SD) scores from 31 patients who reviewed Report A (simple fistula report) and its AI-generated summary, blinded and randomised in order. Subgroup 2: mean scores from 30 patients who reviewed Report B (complex fistula report). Subgroup 3: comparison of scores between AI-generated summaries of Report A and Report B. Paired *t*-tests with *p* values are shown.

**Table 6 diagnostics-16-00072-t006:** Qualitative feedback: what was good and what could be improved?

Domain	Positive Feedback	Points for Improvement
Anatomy	• Correctly explained complex fistula types (e.g., intersphincteric, transsphincteric, supralevatoric) and post-surgical changes.	• Improve anatomical precision by specifying tract height and course through EAS/IAS • Clarify internal opening location using consistent terminology • Correct clockface misapplication (e.g., 7, 9, 11 o’clock confusion) • Avoid vague terms like “back-left” or “muscles around the anus.”
Lay AI-generated summary & Patient-Focused Language	• Provided accessible descriptions of anatomy, healing status, and management, supporting patient understanding.	• Simplify complex anatomical explanations; avoid inferred clinical terms not supported by the radiology report (e.g., active Crohn’s).
Clinical Recommendations	• Suggested suitable next steps (e.g., surgery, EUA, MDT) aligned with imaging findings • Linked findings to red-flag symptoms and possible treatment choices, supporting shared decision-making.	• Avoid minimising complexity of surgery (e.g., labelling major surgery as “minor”) • Ensure impression reflects full extent of disease (e.g., horseshoe tracts) • Make recommendations more specific and tailored to disease severity • Emphasise urgency of MDT review in complex/worsening cases • Refine guidance on seton use and long-term strategies.
Healing & Disease Trajectory	• Recognised the term ‘active infection’ meant intervention was required, whereas the term ‘fibrosis’ meant healing (and therefore observation); recognised blind-ending sinuses and low-risk pockets, supporting cautious optimism or treatment planning.	• Avoid incorrect conclusions (e.g., equating “no proctitis” with “no infection”) • Ensure terminology does not falsely reassure.
Structured Reporting	• Clear and concise summaries, with logical organisation (e.g., old vs. new tracts) • Effective use of clockface notation when accurate • Improved readability via bullet points and spacing • “What should happen next” section praised for clarity.	• Reduce length by removing overly descriptive language • Consolidate repetitive sections (e.g., on next steps) • Avoid over-elaboration that detracts from clarity.
Content Accuracy & Consistency	• –	• Eliminate hallucinated or inferred information not in the original report • Ensure incidental findings (e.g., lymphadenopathy, uterine lesions) are included • Use clinical terms accurately and consistently.
Generic Recommendations	• –	• Avoid generic or unverified assumptions (e.g., presuming UC diagnosis) • Strengthen emphasis on relevant specialist team input (e.g., IBD, colorectal, gynae, nursing) • Suggest appropriate follow-up imaging and surveillance.
Omissions & Missed Findings	• –	• Include all relevant pelvic anatomy (e.g., levator muscles) and fistula measurements • Ensure comprehensive reporting of pelvic/abdominal incidental findings.

**Table 7 diagnostics-16-00072-t007:** Structured Reporting Template.

**Example Structured Radiology Report (Based on Minimum Reporting Dataset—Iqbal et al., 2022 [40])**
• Standard MRI report as per consensus recommendations
B. **Optional MDT-Supportive Section (Clinician-Facing Recommendations)**
**Suggested Next Steps (not directive, for discussion):**
• Further surgical planning needed for complex/horseshoe tracts
• Consider multidisciplinary input (Colorectal Surgeon/IBD Specialist)
• Suggest follow-up MRI in 3–6 months if clinically appropriate
• Raise suspicion for recurrence after prior surgical intervention if present
C. **Lay AI-generated Summary Box: “What This MRI Shows (For Patients)” (Based on ChatGPT-4o output)**
🔍 Your MRI Results Explained
✅ What we found: You have a tunnel (called a fistula) near your back passage. It starts [near the skin/inside the bowel] and may have other branches or small fluid areas.
🔄 Is it healing?: Compared to your last scan, [it looks better/has stayed the same/may need more care].
💬 What does this mean for you?: Your care team might suggest keeping a small soft thread (called a seton) in place, or planning surgery. This depends on your symptoms and type of disease.
If you have Crohn’s disease: This might be part of your condition. Speak to your IBD nurse or colorectal specialist about long-term care.
📅 Next steps: Ask your team about follow-up, treatment options, and whether more scans or procedures are needed.

Note: This is a general summary to help you understand your scan. Always speak to your care team for specific advice.

**Table 8 diagnostics-16-00072-t008:** Example positive recommendations.

“discuss with your colorectal surgeon whether a new seton is needed or if the fistula can be closed”.
Next Steps:
• Although the fistula is small and not infected, it should be reviewed by your colorectal or IBD team.
• Ask your doctor if further treatment is needed, such as antibiotics or minor surgery.
• Monitor and report any new symptoms such as pain, swelling, or discharge.

## Data Availability

The original contributions presented in this study are included in the article. Further inquiries can be directed to the corresponding author.

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
