# Peer review of "Enhancing Patient Understanding of Perianal Fistula MRI Findings Using ChatGPT: A Randomized, Single Centre Study"

_diagnostics, 2025, doi:10.3390/diagnostics16010072_

Round 1
Reviewer 1 Report
Comments and Suggestions for Authors I was glad to review this randomized three-phase single-center study, which investigates the use of a general-purpose Large Language Model (LLMs)(ChatGPT-4o) to generate lay summaries of Magnetic Resonance Imaging (MRI) reports for patients with complex perianal fistulae. The authors concluded that Artificial intelligence (AI)-generated MRI summaries can improve patient communication by enhancing readability, structure, and accessibility, particularly in complex conditions requiring repeated imaging. The manuscript is well-written, and the topic is very interesting. It can be accepted for publication after minor revision: 1) I would suggest adding further information to the title of the study to make the type of manuscript easier to understand. The title could be "Enhancing Patient Understanding of Perianal Fistula MRI Findings Using ChatGPT: A Randomized, Single Center Study." 2) In the abstract section, I would suggest explaining the abbreviations when you first use them For example, MRI (line 18), GPT (line 19), AI (line28), MDT (line 33) 3) In the abstract section, I would suggest adding information about the time period of the study 4) Lines 85-91: The objectives should not be separate. Add this information at the end of the introduction and delete the subtitle "objectives" in line 85. 5) Lines 119 -120: In the materials and methods section, the authors state that exclusion criteria include only malignancy or absence of a fistula. I would suggest adding exclusion criteria in more detail. What about patients <18 years old? 6) I would suggest improving the quality of Figure 1 by adding color as it does not look like a figure currently. 7) " In the last few years, technological developments in the medical field have been rapid and are continuously evolving. One of the most revolutionary breakthroughs was the introduction of the IoT concept within medical practice." Add this important information and make a brief discussion on the role of the Internet of Things and 3D-printing in Patient Understanding of Perianal Fistula MRI Findings 8) In the abstract section, there is no information regarding patients' characteristics such as age, gender, etc. 9) I would suggest adding a figure from an MRI scan showing a transsphincteric fistula in perianal fistulizing Crohn’s disease.Author Response
Reviewer 1:
I was glad to review this randomized three-phase single-center study, which investigates the use of a general-purpose Large Language Model (LLMs)(ChatGPT-4o) to generate lay summaries of Magnetic Resonance Imaging (MRI) reports for patients with complex perianal fistulae. The authors concluded that Artificial intelligence (AI)-generated MRI summaries can improve patient communication by enhancing readability, structure, and accessibility, particularly in complex conditions requiring repeated imaging. The manuscript is well-written, and the topic is very interesting. It can be accepted for publication after minor revision:
- I would suggest adding further information to the title of the study to make the type of manuscript easier to understand. The title could be "Enhancing Patient Understanding of Perianal Fistula MRI Findings Using ChatGPT: A Randomized, Single Center Study."
Thank you for your comment. The title has been updated as follows:
"Enhancing Patient Understanding of Perianal Fistula MRI Findings Using ChatGPT: A Randomized, Single Centre Study." (line 3)
- In the abstract section, I would suggest explaining the abbreviations when you first use them For example, MRI (line 18), GPT (line 19), AI (line28), MDT (line 33)
Thank you for your comment. The following abbreviations have been explained as follows:
Magnetic Resonance Imaging (MRI) (Line 18)
Generative Pre-trained Transformer (GPT-4o) (Line 19)
Artificial Intelligence (AI) (Line 26)
Multi-Disciplinary Team (MDT) (Line 35)
- In the abstract section, I would suggest adding information about the time period of the study
Thank you for your comment. This has been added:
from September 2024 to November 2024 (Line 23)
- Lines 85-91: The objectives should not be separate. Add this information at the end of the introduction and delete the subtitle "objectives" in line 85.
Thank you for your comment. This has been done.
- Lines 119 -120: In the materials and methods section, the authors state that exclusion criteria include only malignancy or absence of a fistula. I would suggest adding exclusion criteria in more detail. What about patients <18 years old?
Thank you for your comment. Exclusion criteria were narrow to ensure a broad capture of as many consecutive scans as possible. The follow has been added.
patients < 18 years (Line 125)
- I would suggest improving the quality of Figure 1 by adding color as it does not look like a figure currently.
Figure 2 (formerly 1) has been updated. The figure has been updated to reflect the original prompt (monochrome).
Figure 2. Final Prompt based on iterative feedback.
- " In the last few years, technological developments in the medical field have been rapid and are continuously evolving. One of the most revolutionary breakthroughs was the introduction of the IoT concept within medical practice." Add this important information and make a brief discussion on the role of the Internet of Things and 3D-printing in Patient Understanding of Perianal Fistula MRI
Thank you for this important comment. Line 487:
The rapid evolution of digital health, particularly the Internet of Things (IoT), may enable real-time symptom tracking and delivery of patient-friendly MRI summaries in the future. Emerging 3D-printing techniques can translate complex fistula anatomy into tangible models to enhance patient understanding and support shared decision-making.
Findings
- In the abstract section, there is no information regarding patients' characteristics such as age, gender, etc.
Thank you for your comment. Line 29:
(mean age 48, 41 % female)
- I would suggest adding a figure from an MRI scan showing a transsphincteric fistula in perianal fistulizing Crohn’s disease.
Thank you for your comment. Figure 1 has been introduced as below.
Line 56: An example MRI of a complex perianal fistula is shown in Figure 1.
Figure 1. Coronal view of a complex transsphincteric fistula with collection in perianal fistulising Crohn’s disease.

Reviewer 2 Report
Comments and Suggestions for Authors
This is a well-conducted and timely study that evaluates the feasibility, accuracy, and patient perception of using a general-purpose large language model (GPT-4o) to generate patient-friendly summaries of MRI reports for perianal fistulas. The manuscript is clearly written, methodologically sound, and addresses an important clinical communication gap. The integration of patient and public involvement (PPI) from the outset is a notable strength. However, several methodological and interpretive aspects could be clarified or expanded to strengthen the manuscript.
1.The abstract mentions GPT-4o, while the Methods section refers to GPT-4-turbo (released November 2023). Please clarify which version was used and ensure consistency throughout the manuscript. If GPT-4o was used in later phases, this should be explicitly stated, along with any implications for performance comparison.
2.The reported hallucination rate of 11.6% is notably higher than some external benchmarks (e.g., ~1.5% for GPT-4o on general tasks). The discussion briefly attributes this to task complexity or stricter clinical scrutiny, but a more nuanced exploration is needed. Consider discussing whether the higher rate reflects the challenge of medical summarization, the specific prompting strategy, or the evaluation criteria used. A comparison with other medical LLM studies (if available) would be helpful.
3.While the final prompt is provided, more detail on the iterative refinement process (e.g., examples of earlier prompts, key changes made based on feedback) would enhance reproducibility. Additionally, the prompt instructs the model to aim for a Flesch-Kincaid score ≥60, but the results show a mean of 65.83. Was this target adjusted during refinement? Please clarify.
4.The classification of “mild harm” (2.4% of cases) is somewhat subjective. Providing concrete examples of what constituted “mild harm” (e.g., specific wording that could cause anxiety or confusion) would help readers assess the clinical significance. Additionally, were any errors considered potentially moderate or severe? If not, please justify.
5.The patient sample in Phase 3 is predominantly White (70.5%) and highly educated (over 57% with university degrees). While this is common in single-center studies, the discussion should more explicitly address how this may limit generalizability to more diverse or lower-health-literacy populations. Consider adding a paragraph on equity and inclusivity implications.
6.Tables 1 and 2 are referenced in the text, but their placement in the submitted draft is somewhat fragmented (e.g., Table 1 split across pages). Ensure all tables are complete and clearly legible in the final version. Also, consider merging Tables 3 and 4 if they are closely related to patient demographics and digital literacy.
7.Ensure consistent use of terms such as “simeticone” vs. “simethicone” (though this appears to be a minor typo in Table 2). Also, standardize the use of “AI-generated summary” vs. “AI summary” throughout.
Author Response
Reviewer 2:
This is a well-conducted and timely study that evaluates the feasibility, accuracy, and patient perception of using a general-purpose large language model (GPT-4o) to generate patient-friendly summaries of MRI reports for perianal fistulas. The manuscript is clearly written, methodologically sound, and addresses an important clinical communication gap. The integration of patient and public involvement (PPI) from the outset is a notable strength. However, several methodological and interpretive aspects could be clarified or expanded to strengthen the manuscript.
1.The abstract mentions GPT-4o, while the Methods section refers to GPT-4-turbo (released November 2023). Please clarify which version was used and ensure consistency throughout the manuscript. If GPT-4o was used in later phases, this should be explicitly stated, along with any implications for performance comparison.
Thank you for your comment. We have clarified and all output were generated using the GPT-4o model. The methods (and any subsequent reference to GPT-turbo) have been updated to reflect this.
Line 113:
The study used GPT-4o (OpenAI, released May 2024), accessed via the ChatGPT Plus interface, to generate patient-friendly summaries from structured radiology report text between February and April 2025
2.The reported hallucination rate of 11.6% is notably higher than some external benchmarks (e.g., ~1.5% for GPT-4o on general tasks). The discussion briefly attributes this to task complexity or stricter clinical scrutiny, but a more nuanced exploration is needed. Consider discussing whether the higher rate reflects the challenge of medical summarization, the specific prompting strategy, or the evaluation criteria used. A comparison with other medical LLM studies (if available) would be helpful.
Thank you for your comment. The discussion has been updated as follows (line 395)
The higher hallucination rate likely reflects the greater difficulty and precision required in medical imaging summarisation, where small anatomical inaccuracies are counted as errors. Hallucination rates in radiology-focused LLM studies are typically higher than in general testing, with one study reporting rates of 6 %40 whilst studies investigating errors in assessment of medical literature have reported rates as high as 40 % 41. It might indicate that creating technically accurate medical summaries is harder than general-domain tasks because there aren’t enough medical reports in the training data, or alternatively that clinical evaluators apply a higher standard of scrutiny than benchmark datasets. Hallucinations increased from the pilot study 42 (no hallucinations) to the full study when prompts involved more complex tasks, such as anatomical localisation using clockface notation. These types of errors, though infrequent, raise safety concerns especially if summaries are delivered without clinical validation. Moreover, the subjective reliability of LLM output is highly sensitive to prompt structure and phrasing, which requires expertise in prompt engineering and introduces further variability.
3.While the final prompt is provided, more detail on the iterative refinement process (e.g., examples of earlier prompts, key changes made based on feedback) would enhance reproducibility. Additionally, the prompt instructs the model to aim for a Flesch-Kincaid score ≥60, but the results show a mean of 65.83. Was this target adjusted during refinement? Please clarify.
Thank you for your comment. The results section has been updated as follows (Line 213):
The iterative refinement of the prompt has been fully described in our original pilot study, published in the Journal of Imaging (PMCID: PMC12471112), which details all earlier prompt versions and the stepwise modifications applied. For the present study, we incorporated both the quantitative results and qualitative patient feedback from that pilot, alongside input from patient representatives within the study group, to develop the final prompt used (Figure 2). Key refinements focused on elements patients identified as most helpful—simple language, clearer anatomical descriptions including clock-face positions, structured actionable recommendations, and improved formatting. The Flesch–Kincaid target of ≥60 was unchanged; the higher mean score observed in the current study (65.83) reflects natural variation rather than a change in the prompt specification.
4.The classification of “mild harm” (2.4% of cases) is somewhat subjective. Providing concrete examples of what constituted “mild harm” (e.g., specific wording that could cause anxiety or confusion) would help readers assess the clinical significance. Additionally, were any errors considered potentially moderate or severe? If not, please justify.
Thank you for your comment. The methods has been expanded: Line 167. Methods
Hallucinations were classified according to the WHO/NHS harm-severity framework, using the International Classification for Patient Safety categories (none, mild, moderate, severe, death) to assess potential patient-impact.
Line 264:
Overall, 2.4 % cases were considered as causing mild harm primarily due to minor discrepancies in anatomical or descriptive details. These errors or hallucinations were considered mildly harmful as they could potentially cause patient anxiety or confusion, e.g. if a summary incorrectly indicated the side of a fistula and, in a worst-case scenario, could mislead clinicians if they relied solely on the AI-generated summary.
Line 269:
No hallucinations were classified as causing moderate or severe harm, as standard clinical practice would require clinicians to verify AI-generated summaries before making any patient care decisions.
5.The patient sample in Phase 3 is predominantly White (70.5%) and highly educated (over 57% with university degrees). While this is common in single-center studies, the discussion should more explicitly address how this may limit generalizability to more diverse or lower-health-literacy populations. Consider adding a paragraph on equity and inclusivity implications.
Thank you for your comment. The limitations has been expanded as follows (line 449):
Although efforts were made to recruit a broad sample of readers, this could be widened further to capture greater diversity. The patient sample in Phase 3 was predominantly White (70.5%) and highly educated (over 57% with university degrees), which may limit the generalizability of findings to more diverse populations or those with lower health literacy. Future studies should aim to include broader, multi-centre cohorts to ensure equity and inclusivity in AI-assisted patient communication research.
6.Tables 1 and 2 are referenced in the text, but their placement in the submitted draft is somewhat fragmented (e.g., Table 1 split across pages). Ensure all tables are complete and clearly legible in the final version. Also, consider merging Tables 3 and 4 if they are closely related to patient demographics and digital literacy.
Thank you for your comment. This has been adjusted to improve readability.
7.Ensure consistent use of terms such as “simeticone” vs. “simethicone” (though this appears to be a minor typo in Table 2). Also, standardize the use of “AI-generated summary” vs. “AI summary” throughout.
Thank you for your comment. “AI-generated summary” has been used throughout the text to standardise terms.
Reviewer 3 Report
Comments and Suggestions for Authors
1. Please explain abbreviations at their first occurrence. For example, what is MRI in the introduction section? We may know this, but it must be clarified for readers who do not. All abbreviations in the manuscript should be reviewed in this manner.
2. The study uses ChatGPT, which is one of the LLM models. How reliable do you think it is to rely on Generative AI models such as ChatGPT?
3. What is the specific reason for using ChatGPT? If such a study was conducted, it would have been beneficial to analyze the results of other GenAI models such as Gemini, DeepSeek, Sider, or Claude as well. These models may produce different outputs from one another. In such a case, which one would be trusted? I am not sure.
4. There are many shortcomings regarding the study’s aim, motivation, and novelty.
5. Clinical evaluation of LLM outputs can be conducted. However, they still need to be reviewed by a specialist. This is because Generative AI produces predictions, and it may make errors in critical tasks such as clinical assessment.
Author Response
Reviewer 3
- Please explain abbreviations at their first occurrence. For example, what is MRI in the introduction section? We may know this, but it must be clarified for readers who do not. All abbreviations in the manuscript should be reviewed in this manner.
Thank you for your comment. The following abbreviations have been explained as follows:
Magnetic Resonance Imaging (MRI) (Line 18)
Generative Pre-trained Transformer (GPT-4o) (Line 19)
Artificial Intelligence (AI) (Line 25)
Multi-Disciplinary Team (MDT) (Line 34)
- The study uses ChatGPT, which is one of the LLM models. How reliable do you think it is to rely on Generative AI models such as ChatGPT?
Thank you for the comment. We have updated our conclusion.
Line 502:
Generative AI models such as ChatGPT can produce coherent and patient-friendly summaries; however, as our study demonstrates, they are prone to inaccuracies (hallucinations) and omissions. Therefore, outputs must be interpreted cautiously and always verified by clinicians before informing patient care. Our study focused on evaluating feasibility, readability, and potential patient comprehension rather than clinical decision-making.
- What is the specific reason for using ChatGPT? If such a study was conducted, it would have been beneficial to analyze the results of other GenAI models such as Gemini, DeepSeek, Sider, or Claude as well. These models may produce different outputs from one another. In such a case, which one would be trusted? I am not sure.
Thank you for your comment. The limitations and discussion section have been updated as follows:
Limitations:
Line 435:
This study did not address several critical systemic issues that were beyond the scope of this feasibility study, such as comparison with competing LLM models (e.g. DeepSeek, Gemini and CoPilot)…
Line 442:
ChatGPT was selected due to its accessibility, widespread use, and robust language generation capabilities at the time of study design. We acknowledge that other LLMs (e.g., Gemini, DeepSeek, Sider, Claude) may produce different outputs, and model performance may vary depending on prompting and domain expertise. Future work could compare multiple models to determine relative reliability, but in clinical contexts, trust must always be mediated by expert review rather than reliance on any single AI model.
- There are many shortcomings regarding the study’s aim, motivation, and novelty.
Thank you for your comment. We have explained the study’s aims on line 92: This randomised feasibility study aims to evaluate the use of a general-purpose LLM (ChatGPT-4o) to generate lay summaries of MRI reports for patients with complex perianal fistulae. The primary objective is to assess patient-perceived comprehensibility, readability, and usefulness. Secondary objectives include a multi-disciplinary clinician evaluation of factual accuracy, completeness, and the presence of hallucinations or misleading content.
This study was designed and developed in conjunction with patient representatives based on the output from a global survey of perianal fistulising Crohn’s disease patients.
We have addressed the limitations of the study from line 434.
- Clinical evaluation of LLM outputs can be conducted. However, they still need to be reviewed by a specialist. This is because Generative AI produces predictions, and it may make errors in critical tasks such as clinical assessment.
Thank you for your comment. We have updated our discussion and conclusion as follows:
Line 446:
in clinical contexts, trust must always be mediated by expert review rather than reliance on any single AI model.
Line 502:
Generative AI models such as ChatGPT can produce coherent and patient-friendly summaries; however, as our study demonstrates, they are prone to inaccuracies (hallucinations) and omissions. Therefore, outputs must be interpreted cautiously and always verified by clinicians before informing patient care. Our study focused on evaluating feasibility, readability, and potential patient comprehension rather than clinical decision-making.
Line 496:
Safe integration requires rigorous clinical oversight, domain-specific model refinement, and ethical safeguards that prioritise patient safety, equity, and trust.
Round 2
Reviewer 2 Report
Comments and Suggestions for Authors
The manuscript can be accepted
Reviewer 3 Report
Comments and Suggestions for Authors
This paper is not suitable for publication in its current form. My decision is reject.